# Mamba®: Vision Mamba Also Needs Registers

## Abstract

Similar to Vision Transformers, this paper identifies artifacts also present within the feature maps of Vision Mamba. These artifacts, corresponding to high-norm tokens emerging in low-information background areas of images, appear much more severe in Vision Mamba—they exist prevalently even with the tiny-sized model and activate extensively across background regions. To mitigate this issue, we follow the prior solution of introducing register tokens into Vision Mamba. To better cope with Mamba blocks' uni-directional inference paradigm, two key modifications are introduced: 1) evenly inserting registers throughout the input token sequence, and 2) recycling registers for final decision predictions. We term this new architecture Mamba®. Qualitative observations suggest, compared to vanilla Vision Mamba, Mamba®'s feature maps appear cleaner and more focused on semantically meaningful regions. Quantitatively, Mamba® attains stronger performance and scales better. For example, on the ImageNet benchmark, our Mamba®-B attains 83.0% accuracy, significantly outperforming Vim-B's 81.8%; furthermore, we provide the first successful scaling to the large model size (*i.e.*, with 341M parameters), attaining a competitive accuracy of 83.6% (84.5% if fine-tuned with $384 \times 384$ inputs). Additional validation on the downstream semantic segmentation task also supports Mamba®'s efficacy.

## 1 Introduction

Recent advances in State Space Models (SSMs) have showcased their considerable potential in sequence modeling tasks. In contrast to Transformers' quadratic computational complexity with respect to sequence lengths, SSMs operate with linear computational complexity, offering significant efficiency advantages in managing extended sequences. One exemplary instantiation of SSMs is the Mamba architecture (Gu & Dao, 2023), which employs selective scan techniques alongside a suite of hardware-optimized designs. This innovation facilitates the efficient training and inference of recurrent models with linear computational complexity and memory overhead. Furthermore, a comprehensive body of recent research (Gu & Dao, 2023; Behrouz et al., 2024; Lieber et al., 2024) substantiates that the Mamba architecture is able to achieve competitive performance levels, on par with Transformers, particularly in processing natural language and audio.

Furthermore, the Mamba architecture has also been successfully extended to a variety of visual tasks (Zhu et al., 2024; Liu et al., 2024b; Li et al., 2024; Hu et al., 2024). The motivation for this expansion mainly arises from the computational challenges presented by processing high-resolution images and videos. These data types often lead to long input sequences that conventional models struggle to handle effectively or efficiently—*e.g.*, for long-length input, traditional models such as Convolutional Neural Networks (CNNs) suffer from relatively small receptive fields, and Vision Transformers (ViTs) contend with high computational and memory costs. Yet, Vision Mamba (Vim) architectures have shown the potential to mitigate these limitations—recent works demonstrate that they not only manage computational and memory demands more efficiently but also deliver strong performance across a variety of generic visual tasks, including classification, segmentation, and image generation (Zhu et al., 2024; Liu et al., 2024a; Hu et al., 2024).

Despite the competitive benchmark performance, our observations reveal that Mamba's internal modeling exhibits significant issues when processing visual inputs. This issue is similar to the prior observation of ViTs (Darcet et al., 2024), where some outlier tokens located in the less semantic background unexpectedly contain rich global information (showing as high attention scores in the feature map). These unusual feature activations are termed artifacts. In this work, we reveal that

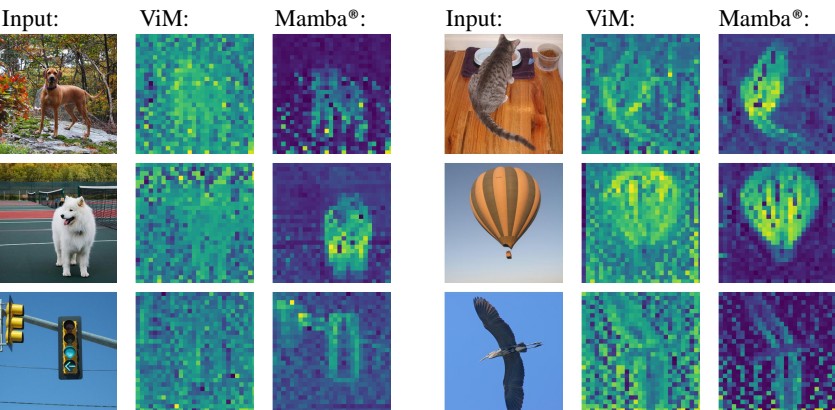

Figure 1: Feature maps of vanilla Vision Mamba (Vim) (Zhu et al., 2024) and our Mamba®. It shows that massive artifacts appear in Vim's feature map, making the model difficult to attend to visually meaningful content within the image. In contrast, our model exhibits much cleaner feature activations, showcasing the significant efficacy of our enhanced architectural design.

this artifact issue not only exists but actually is considerably more severe in Vision Mamba. For example, the artifacts are clearly visible in the feature maps of Vim (Zhu et al., 2024), as illustrated in the 2nd and 4th columns of Figure 1, where activations encompass not just semantically significant content, but also extend to expansive yet minimally informative background regions. Furthermore, as quantitatively confirmed in Section 3.2, these artifact tokens are widely present in different sizes of Vision Mamba models and possess rich global information.

Building upon a previous solution (Darcet et al., 2024), we introduce a straightforward yet effective architectural refinement to Vision Mamba by appending registers—new, input-independent tokens—to the token sequence. Unlike (Darcet et al., 2024) which only appends several register tokens at one end of the input layer, we 1) insert register tokens *evenly* throughout the token sequence; and 2) at the end of the Vision Mamba, concatenate the register tokens to form a comprehensive image representation for the final prediction. We name this enhanced architecture Mamba®.

Empirically, Mamba® showcases advantages on two fronts. Qualitatively, as evidenced by the cleaner feature maps displayed in the 3rd and 6th columns of Figure 1, Mamba® significantly suppresses artifacts, with responses now more focused on visually meaningful content. Meanwhile, as visualized in Figure 6, registers can well capture object-related semantic information for building high-quality image representations. Quantitatively, the improvements in benchmarks are equally compelling. For example, Mamba®-Base achieves an accuracy of 83.0% on ImageNet, notably outperforming the 81.8% accuracy of Vim-Base, which is a vanilla Vision Mamba architecture. Furthermore, Mamba® successfully expands the scaling capabilities of Vision Mamba—whereas previous models were limited to configurations no larger than 90M parameters (Zhu et al., 2024; Liu et al., 2024b; Yang et al., 2024), Mamba® can be effectively trained with up to 341M parameters, reaching an impressive 83.6% accuracy on ImageNet; this accuracy can be further boosted to 84.5% by enlarging the image input size to $384 \times 384$. For the ADE20k (Zhou et al., 2019) semantic segmentation benchmark, our Mamba® attains a 49.1% evaluation mIoU, which significantly outperforms Vim's best result of 44.9% mIoU (Zhu et al., 2024).

## 2 RELATED WORK

**Generic Visual Backbone Architectures.** Modern computer vision predominantly relies on two types of backbone architectures: CNNs that excel in extracting hierarchical features and ViTs that are effective in modeling long-range dependencies. Since the advent of CNNs (LeCun et al., 1998), their structure and scale have undergone a series of significant innovations in recent decades (Krizhevsky et al., 2012; Simonyan & Zisserman, 2015; He et al., 2016; Huang et al., 2017; Tan & Le, 2019; Liu et al., 2022). Unlike CNNs that build spatial dependencies through convolutional operations, ViTs (Dosovitskiy et al., 2021) attain a global receptive field by utilizing the

self-attention mechanism (Vaswani et al., 2017), leading to state-of-the-art performance in a series of downstream visual tasks. Based on this architecture, extensive research has been dedicated to improving its model design (Yuan et al., 2021; Chen et al., 2021a; Liu et al., 2021), enhancing training strategy (Touvron et al., 2021; 2022), and advancing self-supervised pretraining frameworks (Chen et al., 2021b; Caron et al., 2021; Bao et al., 2022; He et al., 2022).

**State Space Models.** The concept of State Space Models (SSMs) can be dated back to the 1960s in control systems where it was used to process continuous inputs (Kalman, 1960). With advancements in discretization strategies (Tallec & Ollivier, 2018; Gu et al., 2020; Nguyen et al., 2022; Gu et al., 2023), SSMs have recently been introduced into the field of deep learning, modeling sequential information such as language and speech (Gu et al., 2022; 2021; Smith et al., 2022). Broadly defined, SSMs can refer to any recurrent models with a latent state such as RNNs and the variant architectures such as Linear Attention (Katharopoulos et al., 2020), RetNet (Sun et al., 2023), and RWKV (Peng et al., 2023). More recently, Gu & Dao (2023) introduced a selective SSM block, namely Mamba, that incorporates structured SSMs with hardware-aware state expansion, leading to a highly efficient recurrent architecture that is competitive to Transformer.

**Mamba Models in Vision.** Building upon the Mamba block, a series of follow-up studies have explored the application of SSMs in computer vision. For example, Zhu et al. (2024) propose a straightforward vision Mamba model by sequentially stacking the Mamba blocks, attaining superior performance than Vision Transformers (Dosovitskiy et al., 2021; Touvron et al., 2021) in both tiny and small model sizes. Liu et al. (2024b) presents a hybrid architecture that combines Mamba with 2D convolution, showcasing significant results in a series of vision tasks. The study of Mamba-based architectures is continuously explored (Lieber et al., 2024; Li et al., 2024; Liu et al., 2024a).

## 3 METHOD

### 3.1 MAMBA PRELIMINARIES

The original definition of SSM is a Linear Time-Invariant (LTI) system that projects the input stimulation $x(t) \in \mathbb{R}^L$ to the output response $y(t) \in \mathbb{R}^L$ through a hidden state $h(t) \in \mathbb{C}^N$. For the continuous inputs, the system can be formulated by a group of linear ordinary differential equations as follows:

$$
\begin{aligned}
h'(t) &= \boldsymbol{A}h(t) + \boldsymbol{B}x(t) \\
y(t) &= \boldsymbol{C}h(t) + Dx(t),
\end{aligned}
\tag{1}
$$

where $\boldsymbol{A} \in \mathbb{C}^{N \times N}$, $\boldsymbol{B} \in \mathbb{C}^N$, $\boldsymbol{C} \in \mathbb{C}^N$, and $D \in \mathbb{C}^1$ denote the weighting parameters.

By discretizing this ordinary differential equation group, the continuous-time SSMs can be integrated to process discrete inputs such as language, speech, and image pixels. To this end, the model can be solved by an analytic solution and then approximated by Zero-Order Hold (Gu & Dao, 2023), leading to a discrete model:

$$
\begin{aligned}
h_t &= \overline{\boldsymbol{A}}h_{t-1} + \overline{\boldsymbol{B}}x_t \\
y_t &= \boldsymbol{C}h_t + Dx_t,
\end{aligned}
\tag{2}
$$

where $\overline{\boldsymbol{A}} = \exp(\Delta A)$, $\overline{\boldsymbol{B}} = (\Delta \boldsymbol{A})^{-1}(\exp(\Delta \boldsymbol{A}) - \boldsymbol{I}) \cdot \Delta \boldsymbol{B}$ are transformed parameters for discrete inputs and $\Delta$ is a learnable parameter estimating the discrete interval. Notably, in contrast to the basic recurrent inference, this structured SSM (S4) allows efficient computation by a convolution process with

$$
\overline{\boldsymbol{K}} = (\boldsymbol{C}\overline{\boldsymbol{B}}, \boldsymbol{C}\overline{\boldsymbol{A}}\overline{\boldsymbol{B}}, \dots, \boldsymbol{C}\overline{\boldsymbol{A}}^{M-1}\overline{\boldsymbol{B}})
\tag{3}
$$

being the kernel and predicting by $\boldsymbol{y} = \boldsymbol{x} * \overline{\boldsymbol{K}}$.

However, the Structural State Space Models' nature of Linear Time-Invariance significantly limits its capacity to fit contextual information, making it difficult to scale up and achieve performance comparable to Transformers. The Selective State Space Model, also known as Mamba or S6 (Gu & Dao, 2023), improves it by introducing input-dependent parameters $\boldsymbol{B} = \boldsymbol{S}_B(x)$, $\boldsymbol{C} = \boldsymbol{S}_C(x)$, and $\Delta = \boldsymbol{S}_\Delta(x)$, leading to a time-varying system that can model more complex inputs. Notably, with associative scan algorithms (Martin & Cundy, 2018; Smith et al., 2022), the Mamba module

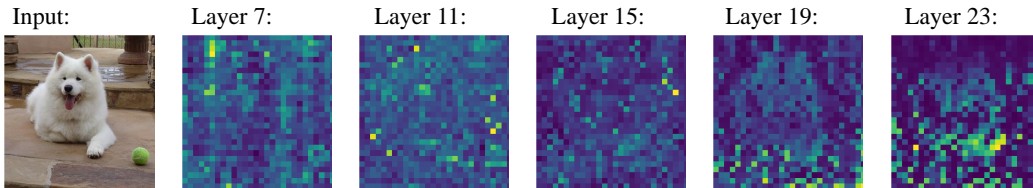

Figure 2: $\ell_2$ norm of local image tokens in vision Mamba's different layers. It shows that massive artifacts associated with high-norm tokens appear in the low-information areas, making it hard to distinguish primary objects from the background.

can be trained and inferred efficiently via parallel computing, with detailed mathematical derivations elaborated in (Gu & Dao, 2023).

To adapt Mamba for visual tasks, images are first processed into sequential inputs through patch embedding as in ViT. However, the standard Mamba is a unidirectional model where each token in the sequence can only access information from preceding tokens. This characteristic, while working well with 1-D language signals, significantly constrains the model's capacity to gather contextual information inherently from 2-D visual signals. To overcome this limitation, a common solution is to reconfigure Mamba blocks for bidirectional scanning. Specifically, the sequence is scanned once from start to end and again from end to start, with the outputs from both scans subsequently averaged to obtain a comprehensive representation (Zhu et al., 2024). We follow this scanning design in all subsequent experiments.

## 3.2 FEATURE ARTIFACTS OF VISION MAMBA

In ViT, an interpretable feature map can be obtained by visualizing the activation scores in their self-attention blocks. Ideally, under appropriate pre-training paradigms, these feature maps are expected to display high attention scores in the informative foreground regions of images and relatively low scores in less semantic background areas. Nevertheless, a considerable amount of outliers often appear in these feature maps, which position-wise correspond to low-information background regions yet exhibit anomalously high attention scores. A recent study (Darcet et al., 2024) has termed these outliers as **feature artifacts**. Specifically, this study reveals that the artifact tokens always possess high normalization values and, during inference, they tend to discard their local information in favor of retaining global features, thereby 'compromising' the quality of the feature map.

This work identifies a similar issue in Vision Mamba models. First, by computing the $\ell_2$ distances between vanilla Vision Mamba's global and local outputs, we observe a considerable amount of activations in background areas (shown in Figure 1). Further analysis of their normalization reveals that these background activations also exhibit high normalization values, akin to the artifacts observed in ViTs. For example, by visualizing the $\ell_2$ normalization of vanilla Vision Mamba's local outputs in Figure 2, we can observe a significant presence of high-norm tokens in the background, even blurring the distinction between foreground and background regions. Quantitatively, we plot the norm distributions of vanilla Vision Mamba in Figure 3a, where it clearly displays a number of outliers with high normalization, confirming consistency with previous findings in ViTs as discussed in (Darcet et al., 2024).

Furthermore, it is equally noteworthy that these artifacts in Vision Mamba function similarly to those in ViTs in retaining global representations (Darcet et al., 2024). As reported in Table 1, the vanilla Vision Mamba model can obtain 81.0% ImageNet accuracy by merely using the average of top 5% high norm tokens as a global feature, which is only 0.1% lower than that of pooling all local tokens. Increasing this threshold to top 10% or 20% high norm tokens further enables the model to match the accuracy of using global pooling. In contrast, relying on the remaining 80% of relatively low-norm tokens results in a performance drop to 79.3%.

*Yet differently*, we observe that the artifact issues are considerably more severe in Vision Mamba than in ViTs: these artifacts appear more prevalent in the background areas and exhibit higher normalization values than those observed in ViTs. As shown in Figure 3a, the average norm of the outlier tokens increases rapidly with the depth of layers, reaching over 4000 by the 23rd layer. Compared

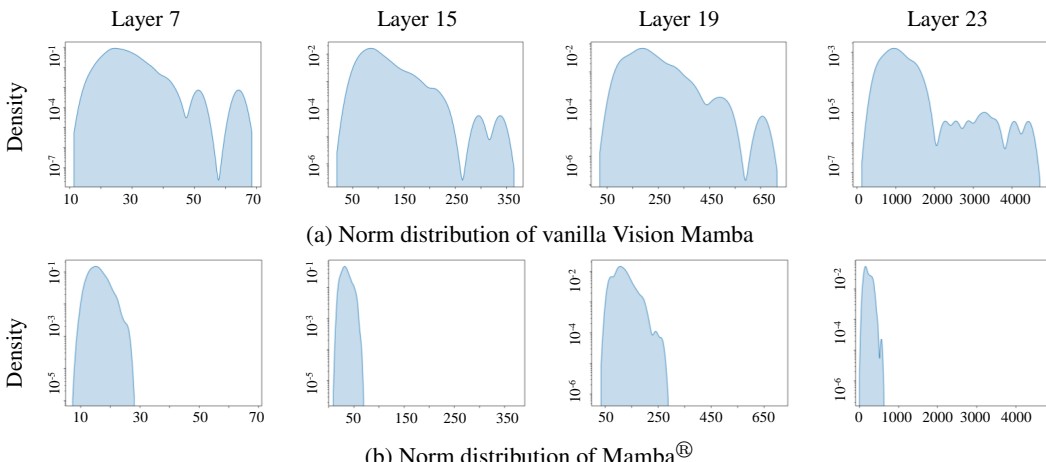

Figure 3: Distributions of $\ell_2$ normalization values of local outputs across different layers. It quantitatively shows that our Mamba® effectively reduces the number of high-norm outliers.

| Feature | Accuracy (%) |
|---|---|
| class token (default) | 81.8 |
| global pooling | 81.1 |
| high-norm tokens (top 20%) | 81.1 |
| high-norm tokens (top 10%) | 81.1 |
| high-norm tokens (top 5%) | 81.0 |
| low-norm tokens | 79.3 |

Table 1: Vim-B's ImageNet accuracy with different features. Using a small portion of high-norm tokens for final prediction attains significantly higher accuracy that that of low-norm tokens.

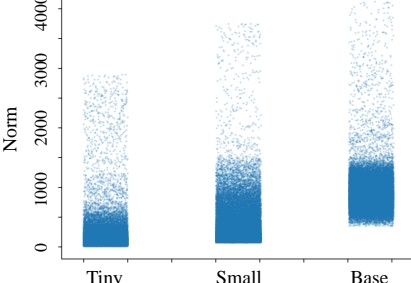

Figure 4: Normalization distribution across different sizes of Vision Mamba.

to the norms below 100 in shallower features, these extremely high-norm artifacts can easily affect feature extraction and pose significant challenges to model optimization, which may potentially explain the instability issues and scaling difficulties encountered in Vision Mamba. Additionally, unlike ViTs where artifacts predominantly appear in larger models, they are present even in the tiny Vision Mamba models and intensify with increasing model size, as illustrated in Figure 4. These observations altogether suggest that the artifact issues are crucial for Vision Mamba models and need to be urgently addressed.

### 3.3 Mamba®: Vision Mamba with Registers

Following the solution of removing artifacts in ViT (Darcet et al., 2024), we propose to address this issue by introducing register tokens into Vision Mamba. We term our new architecture Mamba®. Unlike the previous method (Darcet et al., 2024) which only appends register tokens at one end of the input sequence, we hypothesize that by distributing the register tokens more densely throughout the sequence, our method can 1) better address the more pervasive artifact issue that is unique to vision mamba; and 2) helps capture global representation that is often missed in vision mamba due to its uni-directional nature. The framework of Mamba® is illustrated in Figure 5. Overall, we follow the backbone architecture of vanilla Vision Mamba (Vim) (Zhu et al., 2024), where the input image is first decomposed into a sequence of non-overlapping patches and then fed into a stack of bi-directional Mamba blocks. Based on this plain architecture, we make the following two simple yet very effective modifications to build our Mamba®.

**Sparsely distributed register tokens.** The input sequence of Mamba® is composed of $m$ image tokens produced by patch embedding and $n$ register tokens evenly inserted between them. Contrary

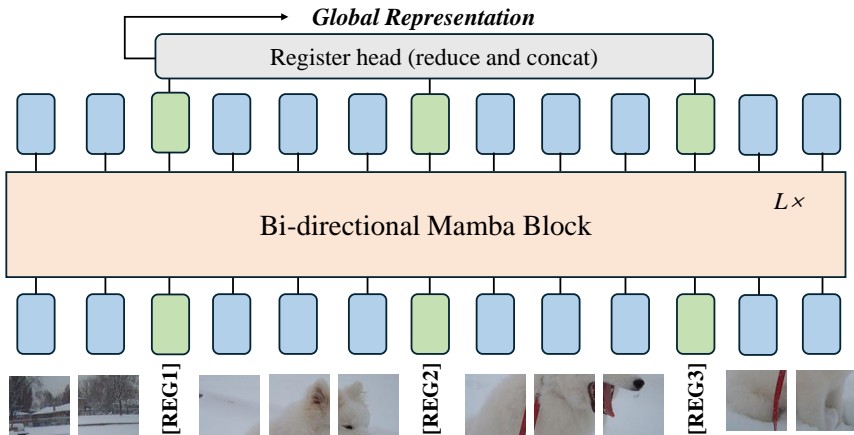

Figure 5: Framework of Mamba®. We address Vision Mamba's artifact issues by evenly inserting input-independent register tokens into the input sequence. In the final layer, we concatenate the output of register tokens to form a global representation for final predictions.

Table 2: Configurations of Mamba®series models. We set patch size to 16 by default for all models.

| Model | Depth | Embed dim ($d$) | #Registers ($n$) | Reduce ($r$) | #Params |
|---|---|---|---|---|---|
| Mamba®-Tiny | 24 | 192 | 12 | 1 | 9M |
| Mamba®-Small | 24 | 384 | 12 | 2 | 28M |
| Mamba®-Base | 24 | 768 | 12 | 4 | 98M |
| Mamba®-Large | 48 | 1024 | 16 | 8 | 340M |

to the self-attention module where token outputs are agnostic to their positions, in Mamba, it is crucial to strategically place the registers to ensure effective interaction with local tokens. Intuitively, for the recurrent Mamba model, sparsely distributed registers facilitate capturing and preserving important semantic information across different positions. In our experiments, we also empirically confirm that this token positioning enhances both quantitative and qualitative performance.

**Register head for final prediction.** Different from ViTs which simply discard registers during the final prediction, we observe that recycling them as a global representation yields significant improvements for vision Mamba. Specifically, given $n$ $d$-dimensional register vectors, we first apply a linear layer to reduce their dimensionality by a factor of $r$, and then concatenate them into a single vector in dimension of $n \times d/r$, which we refer to as the register head. Note that the choice to concatenate, rather than average, is motivated by the multi-head mechanism in self-attention, where concatenation is more effective at retaining information from all heads. The detailed configurations of Mamba®can be found in Table 2.

In addition, as shown in Figure 6, we observe that in certain cases, our registers can interestingly display distinct feature patterns highlighting different objects or semantic elements within a scene, an intriguing aspect that is not explicitly optimized. Given that Mamba currently lacks a multi-head mechanism, this property could have the potential to offer a valuable dimension for interpreting Mamba's feature representations.

## 4 EXPERIMENTS

### 4.1 EXPERIMENTAL SETTINGS

We primarily evaluate our Mamba® on the standard ImageNet (Deng et al., 2009) dataset, which consists of ~1.28 million training images and 50,000 validation images spread across 1,000 categories. Our training setup mostly follows the protocols established in DeiT (Touvron et al., 2021). Specifically, we use AdamW optimizer (Loshchilov & Hutter, 2019) with a momentum of 0.9, a

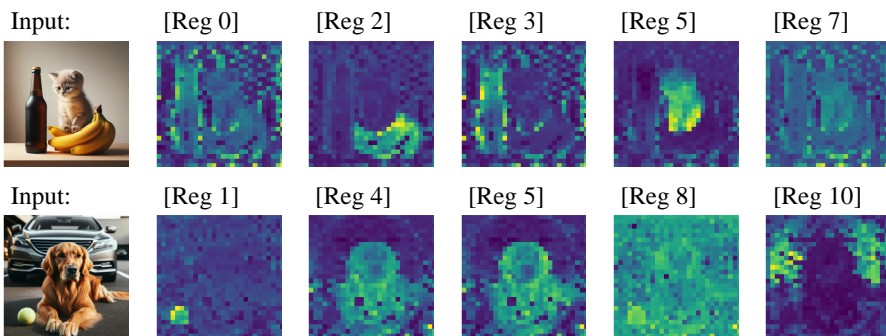

Figure 6: Feature maps for different register tokens. The registers sometimes can attend to different parts or semantics with an image. Similar to the multi-head self-attention mechanism, this property is not required but naturally emerges from training.

Table 3: ImageNet classification results. The throughput is tested on an A100 GPU. The memory overhead is measured with a batch size of 128 on a single GPU. Our results are highlighted in blue.

| Model | Img. size | #Params | Throughput | Mem. | Acc. (%) |
|---|---|---|---|---|---|
| ***Convolutional networks:*** | | | | | |
| ResNet-50 (He et al., 2016) | $224^2$ | 25M | 2388 | 6.6G | 76.2 |
| ResNet-152 (He et al., 2016) | $224^2$ | 60M | 1169 | 12.5G | 78.3 |
| EfficientNet-B3 (Tan & Le, 2019) | $300^2$ | 12M | 546 | 19.7G | 81.6 |
| EfficientNet-B5 (Tan & Le, 2019) | $456^2$ | 30M | 143 | 78.5G | 83.6 |
| EfficientNet-B7 (Tan & Le, 2019) | $560^2$ | 66M | 61 | >80G | 84.3 |
| ConvNeXt-T (Liu et al., 2022) | $224^2$ | 29M | 635 | 8.3G | 82.1 |
| ConvNeXt-S (Liu et al., 2022) | $224^2$ | 50M | 412 | 13.1G | 83.1 |
| ConvNeXt-B (Liu et al., 2022) | $224^2$ | 89M | 305 | 17.9G | 83.8 |
| ***Vision Transformers:*** | | | | | |
| ViT-B/16 (Dosovitskiy et al., 2021) | $384^2$ | 86M | 201 | 63.8G | 77.9 |
| ViT-L/16 (Dosovitskiy et al., 2021) | $384^2$ | 307M | 95 | >80G | 76.5 |
| DeiT-S (Touvron et al., 2021) | $224^2$ | 22M | 1924 | 6.8G | 79.8 |
| DeiT-B (Touvron et al., 2021) | $224^2$ | 86M | 861 | 14.4G | 81.8 |
| DeiT-B (Touvron et al., 2021) | $384^2$ | 86M | 201 | 63.8G | 83.1 |
| ***Hybrid architecture (2D convolution + Mamba):*** | | | | | |
| VMamba-T (Liu et al., 2024b) | $224^2$ | 31M | 464 | 7.6G | 82.5 |
| VMamba-S (Liu et al., 2024b) | $224^2$ | 50M | 313 | 27.6G | 83.6 |
| VMamba-B (Liu et al., 2024b) | $224^2$ | 89M | 246 | 37.1G | 83.9 |
| ***Pure Mamba architecture:*** | | | | | |
| Vim-T (Zhu et al., 2024) | $224^2$ | 7M | 750 | 4.8G | 76.1 |
| Vim-S (Zhu et al., 2024) | $224^2$ | 26M | 395 | 9.4G | 80.5 |
| Mamba®-T | $224^2$ | 9M | 746 | 5.1G | 77.4 |
| Mamba®-S | $224^2$ | 28M | 391 | 9.9G | 81.4 |
| Mamba®-B | $224^2$ | 99M | 196 | 20.3G | 83.0 |
| | $384^2$ | 99M | 63 | 51.4G | **84.3** |
| Mamba®-L | $224^2$ | 341M | 67 | 55.5G | 83.6 |
| | $384^2$ | 342M | 23 | >80G | **84.5** |

weight decay of 0.05, a cosine annealing learning rate starting at $1 \times 10^{-3}$, a batch size of 1024 for Mamba®-Tiny, Small, and Base, and a batch size of 2048 for Mamba®-Large. For better efficiency and preventing over-fitting, we train each model for 300 epochs in 128×128 input size and fine-tune in 224×224 with stronger data augmentation strategies. We also empirically find a 100-epoch intermediate finetuning with weak augmentation can further improve the results. In total, the

Table 4: Semantic segmentation results on ADE20K. All models are trained with an UperNet head and $512\times512$ input size. Our results are highlighted in blue

| Backbone | #Parameters | mIoU (%) |
|---|---|---|
| ResNet-50 | 67M | 41.2 |
| ResNet-101 | 86M | 44.9 |
| DeiT-S | 58M | 43.8 |
| DeiT-B | 144M | 45.5 |
| Vim-Ti | 13M | 41.0 |
| Vim-S | 46M | 44.9 |
| Mamba®-S | 56M | 45.3 |
| Mamba®-B | 132M | 47.7 |
| Mamba®-L | 377M | **49.1** |

Table 5: Ablation study of registers with a Mamba®-Base ($d$=768). The final output dimension is calculated by $d \times n/r$, where $r < 1$ denotes increasing the dimension. Note that the case $n = r = 1$ is equivalent to vanilla Vision Mamba (Vim) with a class token (marked in gray. Our default setup is highlighted in blue. The best result is **bolded**.

| #Reg ($n$) | Rdc ($r$) | Final dim. | Acc. (%) |
|---|---|---|---|
| 1 | 1 | 768 | 81.8 |
| 1 | 1/3 | 2304 | 82.0 |
| 3 | 1 | 2304 | 82.8 |
| 6 | 2 | 2304 | 82.8 |
| 12 | 4 | 2304 | **83.0** |
| 24 | 4 | 4608 | 82.6 |

training process leads to $\sim$230 effective training epochs in $224\times224$ image size, yet significantly outperforms its 300-epoch counterparts. A detailed training recipe can be found in the Appendix.

We further evaluate models' downstream performance on semantic segmentation using the ADE20k (Zhou et al., 2019) dataset, which comprises 150 fine-grained semantic categories distributed across 20K training images, 2K validation images, and 3K test images. Following the existing baseline models (Touvron et al., 2021; Zhu et al., 2024), we choose UperNet (Xiao et al., 2018) as the segmentation head. We utilize AdamW optimizer with a weight decay of 0.01. The models are optimized with a total batch size of 16 for 160k iterations.

## 4.2 MAIN RESULTS

**Image classification.** As illustrated in Table 3, our Mamba® demonstrates strong performance on ImageNet. Compared to the existing pure Mamba architecture, Vim (Zhu et al., 2024), Mamba® shows a significant improvement, outperforming Vim by 1.3% for the Tiny model and by 0.6% for the Small model. More importantly, compared to Vim, our Mamba® exhibits significant enhancement in scalability: we successfully train a Base (99M parameters, achieving 83.0% accuracy) and even a Large (341M parameters, achieving 83.2% accuracy) size Mamba architectures in vision. This performance can be further enhanced by finetuning with the input resolutions increased to $384\times384$—our highest accuracy is 84.5%, which outperforms all prior Mamba variants in ImageNet classification.

**Semantic segmentation.** As shown in Table 4, Mamba® consistently exhibits superior semantic segmentation performance on the ADE20k dataset (Zhou et al., 2019). For example, when compared with Vim-S (Zhu et al., 2024), our Mamba®-S achieves an improvement of 0.4% mIoU. By further scaling up, our Mamba®-B model (featuring 132M parameters) records an mIoU of 47.7%, notably outperforming a similarly-sized DeiT-B model by 2.2% mIoU (results for DeiT are referenced from Liu et al. (2024b)). Additionally, our Mamba®-L (with 377M parameters) also shows great scalability in the segmentation task, achieving 49.1% mIoU on the ADE20k benchmark.

## 4.3 ABLATION STUDY

Basically, in our models, introducing register tokens brings two effects: 1) the inherent benefits of the register itself, including the reduced number of high-norm artifact tokens (see Figure 3b) and enhanced feature extraction capabilities; and 2) the changes in output dimensions caused by the register head (i.e., the $n \times d/r$ output dimension). Here we present ablation studies to separately demonstrate the impact of these two effects on predictive performance.

**Number of registers.** We first ablate how the number of registers affects the model's ImageNet accuracy. As summarized in Table 5, inserting registers generally leads to consistent performance

Table 6: Ablation study of register positions and final prediction protocols. R and I denote register and image tokens respectively. The column "Final prediction" implies how global feature is computed. $R_1$ *only:* use one of the registers and discard others. *"Reduce and concat"* is our default setting that leverages a linear layer to reduce registers' dimension and concatenate them as global representation.

| Mode | Register positions | | | | | | | | Final prediction | Accuracy (%) |
|---|---|---|---|---|---|---|---|---|---|---|
| Head | $R_1$ | $R_2$ | $I_1$ | $I_2$ | $I_3$ | $I_4$ | $I_5$ | $I_6$ | $R_1$ only | 81.3 |
| | | | | | | | | | Mean of registers | 81.4 |
| | | | | | | | | | Reduce and concat | 82.1 |
| Middle | $I_1$ | $I_2$ | $I_3$ | $R_1$ | $R_2$ | $I_4$ | $I_5$ | $I_6$ | $R_1$ only | 81.8 |
| | | | | | | | | | Mean of registers | 82.0 |
| | | | | | | | | | Reduce and concat | 82.6 |
| Even | $I_1$ | $I_2$ | $R_1$ | $I_3$ | $I_4$ | $R_2$ | $I_5$ | $I_6$ | $R_1$ only | 81.7 |
| | | | | | | | | | Mean of registers | 82.2 |
| | | | | | | | | | Reduce and concat | **83.0** |

enhancements, with 0.8% and 1.0% higher accuracy compared with vanilla Mamba architectures in both the Small size and the Base size. Additionally, we observe that simply increasing the output dimension has little benefit to the performance. For example, by projecting Vim-Base's 768-dimensional latent output size into 2304, the accuracy is only improved by 0.1%. Furthermore, we observed that using 12 registers is a sweet point for both the Small size and the Base size; after that, the performance will saturate and may even drop if the final aggregated feature dimension is high (*e.g.*, 4608).

**Registers design choice.** Next, we ablate our design choices of registers, *i.e.*, evenly inserting registers and reuse them for final prediction. The results are reported in Table 6. First, we note the performance is sensitive to the positioning of the registers. For instance, positioning all registers at the beginning of the sequence results in a performance decrease of 0.8% (82.1% *vs.* 83.0%). Similarly, positioning all registers in the middle of the sequence, the best strategy reported by Vim (Zhu et al., 2024), still led to a 0.3% drop in performance, which suggests that the sparse distribution of registers helps with feature extraction for Vision Mamba. These noticeable performance gaps highlight the necessity of evenly inserting registers between image tokens, as Mamba's nature of recurrence makes it sensitive to its token positions in the input sequence.

Further distinctions in register utility are observed when comparing with previous findings (Darcet et al., 2024), which indicate that registers primarily aid in enhancing ViT's feature representation. In contrast, our study demonstrates that the registers play a crucial role in boosting the quantitative performance of Vision Mamba architectures. Notably, utilizing our default method of evenly distributed registers and reusing all for the final prediction achieved an accuracy of 83.0%, surpassing the approach that uses only one register ($R_1$ only; the rest of the tokens are discarded) by 1.2%. These results affirm that registers constitute a vital component of the Vision Mamba architecture.

## 5 CONCLUSION

In this paper, we explored the nature of artifacts within the feature maps of Vision Mamba, contrasting these with those observed in Vision Transformers. Specifically, we implemented a novel architecture named Mamba®, incorporating registers strategically to enhance image processing. Our empirical assessments demonstrate that, qualitatively, Mamba® not only reduces the presence of artifacts but also sharpens the focus on semantically relevant areas within images, leading to cleaner and more effective feature maps. Quantitatively, Mamba® not only surpasses its predecessors in accuracy but also exhibits superior scalability, handling larger model sizes with competitive accuracy. We hope this work can establish a solid backbone architecture for future research in optimizing Mamba architectures in vision.

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

# A  APPENDIX

## A.1  MORE TECHNICAL DETAILS

Table 7: Pre-training configurations

| Configuration | Small | Base | Large |
|---|---|---|---|
| input size | | 128 | |
| epochs | | 300 | |
| optimizer | | AdamW | |
| weight decay | | 0.05 | |
| base learning rate | 5e-4 | 2e-4 | 2e-4 |
| batch size | 1024 | 2048 | 2048 |
| drop path | | 0.1 | |
| label smoothing | | ✗ | |
| random erasing | | ✗ | |
| Rand Augmentation | | ✗ | |
| repeated augmentation | | ✓ | |
| ThreeAugmentation | | ✓ | |

Table 8: Intermediate training configurations

| Configuration | Small | Base | Large |
|---|---|---|---|
| input size | | 224 | |
| epochs | | 100 | |
| optimizer | | AdamW | |
| weight decay | | 0.05 | |
| base learning rate | | 2e-4 | |
| batch size | | 1024 | |
| drop path | 0.2 | 0.4 | 0.4 |
| label smoothing | | ✗ | |
| random erasing | | ✗ | |
| Rand Augmentation | | ✗ | |
| repeated augmentation | | ✓ | |
| ThreeAugmentation | | ✓ | |

Table 9: Fine-tuning configurations

| Configuration | Small | Base | Large |
|---|---|---|---|
| input size | | 224 | |
| epochs | | 20 | |
| optimizer | | AdamW | |
| weight decay | | 0.1 | |
| base learning rate | | 1e-5 | |
| batch size | | 512 | |
| drop path | 0.2 | 0.4 | 0.6 |
| label smoothing | | 0.1 | |
| random erasing | | ✗ | |
| Rand Augmentation | | rand-m9-mstd0.5-inc1 | |
| repeated augmentation | | ✗ | |
| ThreeAugmentation | | ✗ | |

We train Mamba®-Tiny by the configurations of DeiT-Tiny (Touvron et al., 2021) but follow a weaker data augmentation strategy used in (Touvron et al., 2022). For bigger sizes of Mamba®models, we use a three-stage training approach to prevent over-fitting and reduce effective training epochs. We summarize the recipes of pre-training, intermediate training, and fine-tuning in Table 7, Table 8, and Table 9, respectively. For all stages, the learning rate is calculated by

$base\_lr * batchsize/512$, following a cosine decay scheduling with 5 epochs warmup. We use color jitter with a factor of 0.3, mixup and cutmix with alpha setting to 0.8 and 1.0, respectively.

