# OpenReview forum: "Mamba-Reg: Vision Mamba Also Needs Registers"
_ICLR.cc/2025/Conference — ICLR 2025 Conference Withdrawn Submission_

### Official Review · Reviewer_mq1n · 2024-10-30

**Soundness:** 3
**Presentation:** 3
**Contribution:** 1
**Rating:** 3
**Confidence:** 5

**Summary:**

This work investigates the effectiveness of registry tokens, shown to be effective in Vision Transformers (ViTs) [Dercet et al. 2023], for Vision Mamba (ViM) [Zhu et al. 2024]; an extension of the Mamba state space models to vision tasks. The work mainly acts as a continuation of the two preceding works, and shows that registry tokens are beneficial for Vision Mamba models, as previously shown to be the case for Vision Transformers.

Overall, the experiments include higher capacity models than the original work by Zhu et al., and demonstrate the feasibility of registries for ViMs, as previously observed in ViTs. In addition, the overall presentation and motivation is clear. However, there is reason to be critical of the overall novelty and contribution of the work, as the predominant novel contribution is the proposed placement of registry tokens in the sequence. Moreover, the work does not clearly demonstrate any substantial benefits for using a ViM compared to a ViT, other than the inclusion of less than recent baselines (DEiT, Touvron et al. 2021 and ViT, Dosovitskiy et al. 2021). While we appreciate that SSNs are a novel research field, it is of this reviewer's opinion that the work in its current form somewhat misses the opportunity to address the broader context for further research of SSNs and ViMs in vision tasks. In particular, this reviewer would have liked a more clear discussion around the limitations of SSNs in a broader context, particularly to address why (and where) one would choose a ViM as opposed to a ViT.

**Strengths:**

S1: The experiments extend the study of ViMs from Ti-S [Zhu et al. 2024] to more capacities, namely Ti-S-B-L.

S2: The spacing of registry tokens has not been explored for SSNs, and seems like a reasonable ablation given the sequential approach inherent to the proposed method.

S3: The addition of registry tokens to ViMs is shown to have similar benefits to ViTs, which is more or less as expected.

S4: The authors include experiments on dense predictions with segmentation as an additional downstream task.

**Minor Strengths:**
S5: The work parsimoniously connects two previous works in a straightforward manner, and the motivation of the work is; as a result; unambiguous and clear.


**Summary**
The work stands as an earnest contribution to SSMs for vision tasks, particularly for Vision Mamba, which remains a novel research direction for vision tasks. The work connects two existing works, and shows that a largely expected result holds up to experimental validation. Hence, the approach presents a reasonable, simple narrative.

**Weaknesses:**

W1: As the original ViM study [Zhu et al. 2024], the paper exclusively compares the proposed ViM with registries to the original DEiT paper from 2021. This is not the most recent baseline for ViTs trained over IN1k. We believe the authors should instead compare with DEiTv3, which is a more recent work which does end to end pretraining. This omission makes it seem like the proposed ViM with registries are outperforming ViTs, when in fact, this is not the case.

W2: While the proposed study has value for continued study of ViMs, this reviewer finds the novelty of the overall idea behind the study to be limited. ViMs have been shown to approximate the performance of ViTs, and registry tokens have been shown to be effective in ViTs. As such, the results are largely as expected, and there is simply no reason why this architectural optimization technique should be shown to be ineffective in ViMs.

W3: The work largely overlooks what this reviewer believes to be a central question; namely whether ViMs are to be taken as competitive modelling approaches to ViTs. While this is not declared to be within the scope of the paper, it becomes increasingly relevant to investigate how these models hold up to the current de-facto vision backbones to determine the relevancy of the work in the grander context of vision tasks.

W4: The significance of the findings on registry token placement in the sequence, while novel, are somewhat limited in scope.

**Minor Weaknesses:**
W5: The manuscript shows evidence of some unwarranted qualitative hyperbole; e.g., “massive artifacts” (L068). This does not impact the overall score, as it is a stylistic choice.

**Summary**
Unfortunately, this reviewer finds only limited novelty in the overall approach, even after factoring in the effect of placement of registry tokens in the sequence. While the experimental results largely confirms the effectiveness of registry tokens for ViMs, the work acts as a continuation of two existing works, without convincingly arguing for Vision Mamba models as a relevant research goal for the field. This is particularly prevalent from the authors inclusion of baselines. While this choice of baselines goes back to the original work by Zhu et al., the way the results are presented gives the impression that ViMs perform better than ViTs, arguably due to the selection of baselines. In the paper, it is clear that ViMs apply twice the depth of the standard ViT models, and both throughput and memory of all proposed ViMs are poor compared to the corresponding ViTs. These limitations should be more seriously addressed.

**Questions:**

Q1: Both throughput and memory of the proposed models are relatively poor compared to the corresponding ViTs. However, Mamba is presented as being a more efficient alternative to transformers. What factors cause this discrepancy?

Q2: It is not particularly clear why ViMs require twice the depth of standard ViTs (Table 2, L209), nor is it addressed to any extent in the manuscript. Can the authors comment on this?

Q3: Currently, it is unclear why any modeller would currently choose a Vision Mamba model before a Vision Transformer. Given the authors interests in sequence models for vision tasks, can they offer any compelling reason why Vision Mamba should be preferred as a backbone in specific vision tasks?

---

### Official Review · Reviewer_8Xkt · 2024-11-03

**Soundness:** 3
**Presentation:** 3
**Contribution:** 2
**Rating:** 5
**Confidence:** 4

**Summary:**

This paper addresses enhancements to the Vision Mamba (ViM) architecture, a state-space paradigm that has been customized for vision tasks. The primary improvement entails the integration of register tokens into the input sequence, which resolves the severe artifact issue that affects semantic representation in ViM's feature maps. Cleaner feature maps and improved contextual comprehension are enabled by these registers. The enhanced model, known as Mamba®, surpasses its predecessor in critical benchmarks such as ImageNet classification and semantic segmentation on ADE20K, ensuring that efficiency is maintained while improving accuracy and scalability.

**Strengths:**

1. Innovative Application of Registers: The inventive and effective improvement of semantic focus and the mitigation of artifacts in feature maps is achieved through the introduction of register tokens.

2. Improved Performance: Mamba® exhibits a substantial increase in accuracy compared to the base Vision Mamba models, and it is capable of scaling to larger models with competitive results.

3. Scalability and Versatility: The model maintains robust performance across a variety of benchmarks, adapting to varying sizes and duties.

**Weaknesses:**

1. Register Placement Strategy Complexity: Although the paper underscores the advantages of uniformly dispersing register tokens throughout the input sequence, the specific placement strategy could introduce complications in the process of replicating or modifying the architecture.

2. Artifact Sensitivity: Despite the reduction in artifact sensitivity, artifacts continue to present challenges, particularly in deeper layers where high-norm values could potentially impact feature extraction.

3. Ablation Studies with a Limited Scope: The ablation studies that have been presented, while insightful, are limited to the ImageNet dataset. This limitation affects the generalizability of the findings, as a more comprehensive substantiation across multiple datasets is required to bolster the conclusions regarding the efficacy of register tokens and placement strategies. The authors could broaden the scope by conducting similar studies on a variety of datasets, including COCO, ADE20K, or other domain-specific data, to demonstrate the architecture's versatility and robustness in various visual contexts and data distributions.

4. The name of Mamba® causes confusion of the many Mamba family of methods.

**Questions:**

1. Clarification of Register Token Placement Strategy: Could the authors offer a more comprehensive explanation or justification for selecting register tokens that are equitably distributed as the most effective placement strategy? Furthermore, have other configurations, such as adaptive or context-aware placement strategies, been investigated?

2. Ablation Study on Multiple Datasets: I think you can broaden the evaluation to include supplementary datasets, such as COCO, CIFAR-10/100, or ADE20K, in order to verify the proposed method's robustness?

3. Analysis of Persistent Artifacts in Deeper Layers: The paper indicates that high-norm artifacts are still prevalent in deeper layers of the model, albeit to a lesser extent. Could the authors provide further details on the reasons for the persistence of these anomalies at a deeper level, as well as potential mitigation strategies?

4. Comparative Benchmarking with Other SSM-Based Architectures: Although the paper offers comparisons with Vision Transformers and previous versions of Vision Mamba, there is a dearth of discussion regarding the performance of Mamba® in comparison to other state-space models or hybrid SSM architectures that have been modified for vision tasks. Are there any intentions to compare these alternatives or benchmark them?

---

### Official Review · Reviewer_zm6A · 2024-11-03

**Soundness:** 3
**Presentation:** 3
**Contribution:** 3
**Rating:** 5
**Confidence:** 5

**Summary:**

This paper discusses the issue of feature artifacts in Vision Mamba.  Vision Mamba is an unidirectional sequential vision model, consisting of discretized linear transformation layers, which recurrently contextualize the input visual tokens.  Similar to ViTs, such model architectures also have the issue of feature artifacts, where tokens of low-information regions have extremely high normalization values.  Such high-norm tokens are piggy-backed to encode global visual information, sacrificing the performance of downstream dense prediction tasks.  Following prior works, this paper proposes to insert register tokens to carry the global information.  Significant performance gain has been shown on multiple classification and segmentation benchmarks.

**Strengths:**

1. This paper is presented in high quality.  The authors have clearly described their motivation, the context of the problem, and their proposed solution.  The visual examples distinctively show the reduction of feature artifacts with the proposed method.

2. The proposed method is technically sound, which is demonstrated in their experimental results, where the proposed method improves recognition performance without sacrificing much model efficiency.

3. This paper has good analysis of the issue of feature artifact in Vision Mamba.  Based on the results in Figure 3, Table 1 and Figure 4, I am convinced that the issue indeed exists in Vision Mamba.

**Weaknesses:**

1. This paper does not present enough comparison on downstream dense prediction tasks.  The problem of feature artifact is concerning "only" in pixel-wise prediction tasks.  As shown in Table 2 and Table 3 of Darcet et al., 2024, the registered tokens significantly improve the segmentation, depth estimation and unsupervised object discovery task.  However, this paper only shows 0.4% performance gain in segmentation (ViM-S vs. Mamba-S).  It's unclear if the proposed method is as effective in other dense prediction tasks.

2. This idea of having registered tokens in Mamba should also apply to Hybrid architectures.  This paper doesn't show if the idea works on VMamba models.

**Questions:**

1. Could the proposed method improve other dense prediction tasks, including depth estimation and unsupervised object discovery?

2. What is the performance of Vim-B and Vim-L in semantic segmentation on ADE20K? Does the proposed method improve Vim at these scales?

3. Does the proposed method work on hybrid architectures?

----

I would increase my scores if the authors could answer the above questions during rebuttal.

---

### Official Review · Reviewer_h9wX · 2024-11-04

**Soundness:** 2
**Presentation:** 3
**Contribution:** 1
**Rating:** 3
**Confidence:** 4

**Summary:**

The paper identifies that high-norm tokens emerging in low-information backgrounds is a more pronounced issue in Vision Mamba. To address this, the paper introduces Mamba-Reg, which inserts registers evenly throughout the input token sequence. In the final prediction, Mamba-Reg recycles these registers as a global representation. Experiments on ImageNet and ADE20K are conducted to evaluate the model’s performance.

**Strengths:**

1. The concept of evenly inserting registers and recycling them for prediction is novel.
2. Figure 6 demonstrates that the register mechanism effectively distinguishes different parts of the image.
3. The paper is well-written and easy to follow.

**Weaknesses:**

1. **Lack of Theoretical Analysis**: The paper lacks a thorough theoretical analysis explaining why evenly distributed registers resolve the high-norm token issue. This weakens the connection between the high-norm token analysis in Vision Mamba and the introduction of register tokens. Additionally, there is insufficient analysis on the norm behavior after introducing the register tokens.

2. **Insufficient Experimentation**: In Table 3, the comparison omits Vision Mamba at both the Base and Large scales, making it difficult to assess performance in larger, deeper models. The overall performance of Mamba-Reg also appears to be weaker than Vision Mamba.

3. **Limited Dataset for Validation**: ImageNet alone is insufficient to validate performance. Additional experiments on ImageNetV2 are recommended to determine whether the observed performance gains are due to overfitting.

4. **Incomplete Semantic Segmentation Comparison**: The comparison in semantic segmentation is inadequate, with only parameter counts provided. Key metrics such as FLOPs, throughput, and memory usage are missing. Results on alternative architectures, such as Mask2Former, should also be included. Additionally, the Mamba-Reg T results are missing.

5. **Additional Downstream Tasks Needed**: Further downstream tasks, such as object detection and instance segmentation, should be included to comprehensively demonstrate the model's applicability.

6. **Limited Differentiation Between Middle and Even Insert**: The performance difference between middle and even insert methods on ImageNet classification is minimal. Conduct ablation studies on additional vision tasks to more clearly demonstrate the advantage of even insertion.

**Questions:**

See weakness

---

### Official Review · Reviewer_x3zz · 2024-11-06

**Soundness:** 3
**Presentation:** 3
**Contribution:** 2
**Rating:** 6
**Confidence:** 3

**Summary:**

This paper addresses the issue of artifacts within the feature maps of Vision Mamba. Similar to vision transformers, high-norm tokens emerging in low-information background areas of images. To mitigate this, the authors introduce register tokens into Vision Mamba, resulting in a new architecture termed Mamba-R. This architecture includes two key modifications: 1) evenly inserting registers throughout the input token sequence, and 2) recycling registers for final decision predictions. Qualitative observations suggest that Mamba-R's feature maps are cleaner and more focused on semantically meaningful regions. Additional validation on downstream semantic segmentation tasks further supports Mamba-R's efficacy.

**Strengths:**

1. The paper is well-written and presents its ideas in a clear and understandable manner.
2. The exploration of alleviating artifacts in Vision Mamba is a valuable contribution to the field.
3. The experiments provide evidence supporting the necessity of register tokens in Vision Mamba.

**Weaknesses:**

1. The paper primarily transfers observations and methods from Vision Transformers [1] to Vision Mamba without significant innovation.
2. Despite the introduction of registration, the feature maps still exhibit noise compared to visualizations in prior work [1], why?
3. Mamba-R introduces additional parameters and computations, as noted in Table 4, where a small improvement in mIOU (0.4) comes at the cost of 10M more parameters.
4. The authors are encouraged to validate the effectiveness of registration in unsupervised object discovery, similar to the analysis in Table 3 of [1].
5. The strategy of distributing register tokens evenly throughout the sequence appears less convincing, as indicated by marginal improvements (only +0.3 in Table 6) compared to puting registration tokens in the middle. This could be attributed to randomness rather than a systematic advantage.
6. Generalization to other representative vision mambas, like VMamba [2] and MambaOut [3]?

[1] Vision Transformers Need Registers
[2] VMamba: Visual State Space Model
[3] MambaOut: Do We Really Need Mamba for Vision?

**Questions:**

Please refer to weaknesses.

---

### Note · Authors · 2024-11-12

I have read and agree with the venue's withdrawal policy on behalf of myself and my co-authors.